# OTX015 Epi-Drug Exerts Antitumor Effects in Ovarian Cancer Cells by Blocking GNL3-Mediated Radioresistance Mechanisms: Cellular, Molecular and Computational Evidence

**DOI:** 10.3390/cancers13071519

**Published:** 2021-03-25

**Authors:** Francesca Megiorni, Simona Camero, Paola Pontecorvi, Lucrezia Camicia, Francesco Marampon, Simona Ceccarelli, Eleni Anastasiadou, Nicola Bernabò, Giorgia Perniola, Antonio Pizzuti, Pierluigi Benedetti Panici, Vincenzo Tombolini, Cinzia Marchese

**Affiliations:** 1Department of Experimental Medicine, Sapienza University of Rome, Viale Regina Elena 324, 00161 Rome, Italy; paola.pontecorvi@uniroma1.it (P.P.); simona.ceccarelli@uniroma1.it (S.C.); eleni.anastasiadou@uniroma1.it (E.A.); antonio.pizzuti@uniroma1.it (A.P.); cinzia.marchese@uniroma1.it (C.M.); 2Department of Maternal and Child Health and Urological Sciences, Sapienza University of Rome, Viale Regina Elena 324, 00161 Rome, Italy; simona.camero@uniroma1.it (S.C.); camicia.1630843@studenti.uniroma1.it (L.C.); giorgia.perniola@uniroma1.it (G.P.); pierluigi.benedettipanici@uniroma1.it (P.B.P.); 3Department of Radiological, Oncological and Pathological Sciences, Sapienza University of Rome, Viale Regina Elena 324, 00161 Rome, Italy; francesco.marampon@uniroma1.it (F.M.); vincenzo.tombolini@uniroma1.it (V.T.); 4Faculty of Bioscience and Technology for Food, Agriculture and Environment, University of Teramo, 64100 Teramo, Italy; nbernabo@unite.it

**Keywords:** BET inhibitors, OTX015, ovarian cancer, GNL3 (nucleostemin), radioresistance

## Abstract

**Simple Summary:**

The outcome for women diagnosed with ovarian cancer (OC), the most aggressive gynecological tumor worldwide, remains very poor. Encouraging therapeutic impact of epigenetic drugs has been suggested in a wide range of human solid tumors, including OC. The present study assessed the in vitro cytostatic and cytotoxic effects of OTX015, a pan Bromodomain and Extra-Terminal motif inhibitor, in human OC cells, both as single treatment and in combination with radiotherapy. Cellular, molecular and computational network analyses indicated the centrality of GNL3 downregulation in mediating the OTX015-related antitumor efficacy that blocks disease progression/maintenance and radioresistance acquisition. Our preclinical results confirm that targeted and combinatorial treatments represent effective anticancer strategies to be translated in the clinical research for improving OC patient care.

**Abstract:**

Ovarian cancer (OC) is the most aggressive gynecological tumor worldwide and, notwithstanding the increment in conventional treatments, many resistance mechanisms arise, this leading to cure failure and patient death. So, the use of novel adjuvant drugs able to counteract these pathways is urgently needed to improve patient overall survival. A growing interest is focused on epigenetic drugs for cancer therapy, such as Bromodomain and Extra-Terminal motif inhibitors (BETi). Here, we investigate the antitumor effects of OTX015, a novel BETi, as a single agent or in combination with ionizing radiation (IR) in OC cellular models. OTX015 treatment significantly reduced tumor cell proliferation by triggering cell cycle arrest and apoptosis that were linked to nucleolar stress and DNA damage. OTX015 impaired migration capacity and potentiated IR effects by reducing the expression of different drivers of cancer resistance mechanisms, including GNL3 gene, whose expression was found to be significantly higher in OC biopsies than in normal ovarian tissues. Gene specific knocking down and computational network analysis confirmed the centrality of GNL3 in OTX015-mediated OC antitumor effects. Altogether, our findings suggest OTX015 as an effective option to improve therapeutic strategies and overcome the development of resistant cancer cells in patients with OC.

## 1. Introduction

Ovarian cancer (OC) is a rare but highly lethal gynecologic malignancy, representing 3% of all tumors and the fifth leading cause of cancer mortality in women worldwide [1]. This high death-to-incidence ratio is essentially due to the absence of OC-related specific symptoms and the lack of effective screening strategies, so many women are diagnosed at advanced stage of the disease, when cancer has already spread into the abdominal cavity [2]. Although conventional therapies, including cytoreduction surgery and chemo/radiotherapy, are the first-line treatments for OC tumors, a high percentage of patients can present a rapid progressive disease or severe relapse after treatment ends due to the appearance of resistant cancer cells [3]. Indeed, the overall prognosis of patients diagnosed with OC remains poor, with a 5-year event-free survival rate of about 50% [4]. So, the identification of novel sensitizing strategies, mainly addressed to overcome tumor resistance and recurrence, is urgently required in the treatment of patients with OC.

Genetic predisposition accounts for more than 50% of OC tumors, mainly associated with inherited mutations in the *BRCA1*, *BRCA2* or *TP53* genes, which are essentially related to genomic instability [5]. Furthermore, epigenetic modifications, which include DNA methylation, histone modifications, and post-transcriptional gene regulation by non-coding RNAs (ncRNAs), have increasingly been associated with the development, progression and response to therapies of different human cancers, including OC [6]. Epigenetic alterations, primarily involving DNA methylation and histone modifications (i.e., acetylation, methylation and phosphorylation), are able to regulate gene expression by altering chromatin structure and DNA accessibility to transcriptional factors without modifying nucleotide sequence. One of the most important epigenetic modification is represented by histone acetylation, which is dynamically modulated by histone acetyltransferases (HATs) and deacetylases (HDACs) as well as by the action of the Bromodomain and Extra Terminal (BET) family proteins (BRD2, BRD3, BRD4 and the germ cell-expressed BRDT), that are able to read the acetylated lysines and control the recruitment of transcriptional machinery at the specific regulatory regions of a gene for starting and maintaining its expression. Different studies have established that epigenetic imbalance is implicated in cancer disease [7,8], comprehending OC [6,9]. Indeed, aberrant expression of BRD4 has been frequently found in OC patients and it has been correlated with poor prognosis [10].

Considering the pivotal role of epigenetic drivers in the regulation of gene expression, many drug inhibitors have been recently developed and tested against these promising targets. In particular, different BET inhibitors (BETi), which have a significant impact on BRD2, BRD3 and BRD4 levels and activity, have been assessed to exert antitumor activity against a set of hematological malignancies and solid tumors [11,12,13,14]. A novel and selective BETi, the thienotriazolodiazepine compound OTX015, is now available for human use [15]. OTX015 competitively occupies the acetyl-binding pockets of BRD2, BRD3 and BRD4 proteins, this resulting in their release from active chromatin with consequent suppression of transcription. This molecule has shown to have potent antitumor effects in in vitro preclinical studies, is orally bioavailable and has favorable pharmacokinetic properties in murine cancer models [16,17,18]. OTX015 has recently entered phase 1/2 clinical trials, comprehending different hematologic malignancies, selected advanced solid tumors and glioblastoma multiforme [12,14,19]. Notwithstanding the information about the anticancer effects of different BETi agents, mainly JQ1 drug, in OC [20,21,22,23], it is useful to investigate their molecular mechanisms and their precise targeted factors both as monotherapies and combined with conventional treatments. Our recent observations have linked OTX015 antitumor activity to a marked downregulation of *GNL3* gene [11], coding for the G nucleolar 3 protein (often indicated as nucleostemin) that is implied in growth, survival, genome stability and stemness capacities of various cancer cell types [24], this being also related with cancer resistance to conventional treatments [24,25,26,27,28]. Currently, only few studies have correlated the aberrant expression of GNL3 gene with OC tumorigenesis [29,30].

In the present study, we focused on the molecular mechanisms involved in the antitumor effects exerted by OTX015 in OC cells by demonstrating that BET inhibition reduces cancer cell proliferation and viability through the marked downregulation of GNL3 expression. The functional role of the *GNL3* gene product was assessed by its specific knocking-down in OC cells by RNA interference strategy. By interrogating large available databases from OC tumors, we showed higher *GNL3* transcript levels in OC patients than in healthy ovarian tissues. Interestingly, we also applied a computational approach, based on Barabasi-Albert (BA) modelling [31,32], to define the GNL3-related network architecture in which connected components are represented by key molecules in cancer biology. Finally, OTX015 exposure increased OC cell sensitivity to radiation therapy by keeping GNL3 expression at low levels, this indicating that the combination of epi-drugs and standard treatments might be a favorable clinical approach to potentiate antitumor efficacy and reduce the onset of resistant cancer cells. Altogether, our results provide a rational for the use of BETi as an adjuvant therapeutic approach to improve clinical outcome of patients with OC and suggest GNL3 to have a prognostic significance in evaluating OC clinical parameters.

## 2. Materials and Methods

### 2.1. Tumor Samples

OC primary tumor biopsies were obtained before any chemotherapeutic treatment from *n*. 22 patients (median age 57 years; range 32–89 years) who underwent surgical procedure at the to the Department of Maternal, Infantile and Urological Sciences, Sapienza University of Rome (Rome, Italy). Of the 22 serous carcinomas, 1 were classified as International Federation of Gynecology and Obstetrics (FIGO) stage II, 19 as stage III, and 2 as stage IV. With regard to the histologic grade, two were G1 and 20 were G3. Five OC patients carried *BRCA1* mutations. Normal ovarian tissues, obtained from *n* = 8 patients (median age 56 years; range 47–69 years) without oncological diseases, served as negative controls. All specimens were snap-frozen and stored at −80 °C until molecular analysis was performed. Informed consent was obtained from all participants included in this study. 

### 2.2. Reagents and Irradiation

OTX015 (MK-8628) was supplied as lyophilized powder by Selleckchem (Suffolk, UK) and reconstituted in dimethyl sulfoxide (DMSO, Sigma, St. Louis, MO, USA). Control cells were treated with DMSO at the maximum amount used to deliver OTX015. Irradiation was carried out at 4 Gy (380 UM/min) at the Department of Radiological, Oncological and Pathological Sciences, Sapienza University of Rome (Rome, Italy), as previously described [11].

### 2.3. Cell Lines, Cultures in Adherent Conditions

Established OC cell lines SKOV3 and UWB1.289 were purchased from the American Type Culture Collection (ATCC; Manassas, VA, USA). SKOV3 cells (*BRCA1* wild type) were grown in RPMI-1640 medium (Sigma-Aldrich; Merck KGaA, Darmstadt, Germany) supplemented with 10% fetal bovine serum (FBS, Sigma-Aldrich), 2 mM L-glutamine, 100 µg/mL streptomycin and 100 U/mL penicillin; UWB1.289 cell line (*BRCA1*-null) was cultured in 50% RPMI and 50% MEGM (Mammary Epithelial Growth Medium), supplemented with 3% FBS, 2 mM L-glutamine, 100 µg/mL streptomycin and 100 U/mL penicillin. To minimize the risk of working with misidentified and/or contaminated cells lines, cells were used at low passages (<30 subcultures) and frequently tested for mycoplasma contamination.

### 2.4. Cell Proliferation Assays and Morphological Assessment of OTX015-Treated Cells

Direct counting of SKOV3 and UWB1.289 living cells was performed by trypan blue exclusion dye (Sigma-Aldrich, Saint Louis, MO, USA) using the Countess II Automated Cell Counter (Thermo Fisher, Waltham, MA, USA) after OTX015 treatment (0–1–3 µM) for 72 h. Experiments were repeated at least two times, with each reading in duplicate. An Axio Vert.A1 microscope (Carl Zeiss Microscopy, Thornwood, NY, USA), furnished with an AxioCam MRc5 camera (Carl Zeiss Microscopy) was used to observe the morphological changes of the cells treated or untreated with OTX015 (1 µM and 3 µM). After being cultured for 72 h, cells were stained by the standard Giemsa method, as previously reported [32], and photographed under the light microscope, using a 20× magnification.

### 2.5. Spheroid Formation

SKOV3 cells (2 × 10^4^) were plated into ultra-low attachment 24-well culture plates (Corning, NY, USA) with an appropriate volume (500 μL) of serum-free medium composed by DMEM/F12 (Invitrogen, Carlsbad, CA, USA), 1× B-27 Supplement (Invitrogen), 20 ng/mL basic fibroblast growth factor (FGF, Peprotech, Rocky Hill, NJ, USA) and 20 ng/mL epidermal growth factor (EGF, Peprotech). After 48 h, spheroid formation was checked and cells were treated with OTX015. Sphere-forming efficiency was monitored under a light microscope at 4, 7 and 10 days after drug-treatment. DMSO was used in control cells.

### 2.6. RNA Extraction and Quantitative Real Time PCR (q-PCR)

Tissue samples (OC and normal ovary) were disrupted for subsequent RNA extraction by using the TissueLyser II instrument (Qiagen, Hilden, DE, USA). Total RNA was extracted using Trizol (Invitrogen) and reverse transcription was carried out with SensiFAST cDNA Synthesis Kit (Bioline, London, UK). SYBR Green quantitative Real Time PCR (q-PCR) experiments for human *BRD4*, *GNL3* and *MYC* transcripts were carried out with the SensiFAST SYBR Hi-ROX Master Mix (Bioline). Primer sequences are available under request. Samples were normalized according to the *β-actin* mRNA levels. All q-PCR experiments were performed on a StepOne Real Time System (Applied Biosystems, Foster City, CA, USA) machine. Relative mRNA levels were obtained by the comparative Ct method, as previously described [11,33]. Samples were run in triplicate in at least two different experiments.

### 2.7. Cell Cycle Analysis

OC cells exposed to OTX015 (0–1–3 μM) for 48 h were collected and fixed in ice-cold ethanol. Control cells were treated with DMSO. Cell cycle distribution was analyzed by Propidium Iodide (PI, Sigma-Aldrich; Merck KGaA) staining by BD FACSCalibur (BD Biosciences, Franklin Lakes, NJ, USA), as previously described [11]. Data were analyzed with ModFit 3.1 software (Verity Software House, Topsham, ME, USA). Two independent experiments were performed for each treatment.

### 2.8. Apoptosis Analysis

Apoptosis was analyzed by flow cytometry using the PE-Annexin V Apoptosis Detection Kit I (BD Biosciences), as previously described [11]. Briefly, OC cells were seeded overnight in 6-well plate and then treated with different doses of OTX015 (0–1–3 μM) for 72 h. Control cells were treated with DMSO. PE-Annexin V and 7-AAD fluorescence intensities of control or treated samples were analyzed using a BD FACSCalibur (BD Biosciences) with the Cell Quest Pro software (BD Biosciences) in order to detect early apoptotic (Annexin V positive/7-AAD negative) and late apoptotic (Annexin V positive/7-AAD positive) cells. Two independent experiments were performed for each treatment.

### 2.9. Colony Formation Assay

For anchorage-dependent colony formation assays, OC cells treated or not with OTX015 were plated in triplicate in 6-well plates at 2 × 10^3^ cells/well and cultured for 12 days. Colonies were fixed in 100% methanol, stained with 0.1% crystal violet dye in methanol and photographed by ChemiDoc XRS+ (Bio-Rad, Hercules, CA, USA). Colony quantification was performed by crystal violet solubilization as previously described [34]. Assays were carried out at least in two independent experiments.

### 2.10. Migration Assays

OC cells were cultured in complete medium with OTX015 (0–1–3 μM) for 48 h before plating 8 × 10^4^ cells per well into BD FalconTM Cell Culture Inserts with 8 μm pore polycarbonate filters (Falcon), as previously described [11]. Cells were photographed under a light microscope at 10× or 20× magnifications; 8 randomly selected fields were examined and counted manually. The average number of migrated cells was calculated. Experiments were performed in triplicate and repeated twice.

### 2.11. Protein Extracts and Western Blot Analysis

For Western blotting (WB) experiments, whole protein extracts were prepared using RIPA buffer, as previously described [11]. Filters were hybridized with the following primary antibodies: ATM, BCL2, DNA-PK, MYC, p21 and PTEN by Santa Cruz Biotechnology (Dallas, TX, USA); NRF2 and GNL3 by Abcam, Cambridge, UK; γH2AX, phospho(p)-AKT, AKT, cleaved caspase-3 and cleaved PARP by Cell Signaling Technology, Danvers, MA, USA; BRD4 by Bethyl; CD133 by ThermoFisher Scientific, Waltham, MA, USA. Appropriate horseradish peroxidase (HRP)-conjugated secondary antibodies (Bethyl, Montgomery, TX, USA) were used. Protein signals were detected using WesternBright ECL kit (Advansta, Menlo Park, CA, USA), and visualized by ChemiDoc XRS+ (Bio-Rad). Tubulin (Santa Cruz Biotechnology) was used as a normalization control for equal loading of total proteins and cytoplasmatic fraction in SKOV3 cells. β-actin (Sigma) served as loading control for equal loading in UWB1.289 cells. Lamin B (Santa Cruz Biotechnology) was used as loading control for nuclear fraction. Densitometric analysis was performed using the Image Lab 5.1 software (Bio-Rad), as already reported [11].

### 2.12. Immunofluorescence

Immunofluorescence (IF) experiments were performed as previously described [35]. Briefly, 6 × 10^4^ SKOV3 cells were plated in 24-well plates with 2% gelatin-coated glasses and treated with OTX015 (1 μM and 3 μM). DMSO was used in control cells. After 48 h, cells were fixed in 4% paraformaldehyde, permeabilized in 0.1% Triton X-100 and incubated with the following primary antibodies: BRD4 (Bethyl); GNL3 (Abcam); γH2AX (Cell Signalling Technology, Danvers, MA, USA); MYC (Santa Cruz Biotechnology). Texas Red-conjugated secondary antibodies (Jackson ImmunoResearch, West Grove, PA, USA) were used and nuclei were counterstained with DAPI (Sigma-Aldrich). IF signals were captured by a Zeiss ApoTome (Carl Zeiss, LLC, White Plains, NY, USA) using a 40× magnification. Images were acquired by Axiovision software (Carl Zeiss). Experiments were replicated twice.

### 2.13. Transient Transfection

SKOV3 cells (10^5^ cells/well in 12-well plates) were transfected with a pool of small interfering RNAs (siRNAs) against human *BRD4* (si-BRD4, sc-43639 by Santa Cruz Biotechnology), *GNL3* (si-GNL3, sc-45830 by Santa Cruz Biotechnology), or siRNA negative control (si-NC, sc-37007 by Santa Cruz Biotechnology) by using RNAiMAX (Invitrogen), as previously indicated [11]. Specific gene knocking-down was assessed after transfection at 72 h by q-PCR and WB. The PTEN-vector (Plasmid #78776), purchased from Addgene (Cambridge, MA, USA), was delivered into SKOV3 cells by Lipofectamine 2000 (Invitrogen). Cells transfected with empty plasmid (pcDNA3.1) were set as the negative control. PTEN upregulation-mediated effects were tested by WB at 72 h after transfection.

### 2.14. Statistical Analysis

All analyses were performed using GraphPad Prism 7.0 software for Mac (GraphPad Software, La Jolla, CA, USA). Results are presented as mean ± standard deviation (SD) of each condition. Statistical significance between groups was assessed by two tailed independent Student’s *t*-test or ANOVA comparisons and probability (*p*) values of less than 0.05 were accepted as significant (* *p* < 0.05; ** *p* < 0.01; *** *p* < 0.001). All the experiments were done in triplicates and repeated three times unless mentioned otherwise.

### 2.15. Computational Analysis

*GNL3* gene expression analysis across public OC datasets was obtained by interrogating the R^2^ Genomics and Visualization Platform (http://r2.amc.nl (accessed on 20 December 2020)) from the website. In particular, MegaSampler algorithm was used and expression graphs of *GNL3* in different OC cohorts were downloaded and formatted for publication. To explore the functional link between *GNL3* expression and the DNA repair as well as apoptotic pathways, we used Cytoscape 3.6.0. We imported the data referring DNA repair-related genes, apoptosis-related genes and *GNL3* from IntAct Molecular Interaction Database (https://www.ebi.ac.uk/intact/ (accessed on 20 December 2020)), an on-line archive that provides a freely available, open-source database system and analysis tool for molecular interaction data. All interactions are derived from literature curation or direct user submissions and are freely available. The data were accessed on 20 December 2020. We merged the obtained network, thus originating the GNL3 Network GNL3_N and then, by using the Network Analyzer plug-in, we computed the topological parameters listed in Appendix A. The main topological parameters assessed in this study, specifically the identification of MC_GNL3_N hubs and of BottleNecks nodes, were performed as reported in Appendix A.

## 3. Results

### 3.1. OTX015 Inhibits Cell Viability and Proliferation of OC Cell Lines

The antitumor effects of OTX015, a new pan BET inhibitor used in 1/2 phase clinical trials in different malignancies, have been assessed in ovarian cancer (OC) in vitro models, SKOV3 cells, representing an adenocarcinoma OC tumor, and UWB1.289 cells, deriving from a papillary serous histology OC tumor. Cancer cell lines were exposed for 72 h to increasing concentrations of the epi-drug, with values ranging from 0 to 20 μM, and cellular viability was evaluated by MTT assays (data not shown). Proliferation and viability of OC cells were significantly inhibited by OTX015 exposure in a dose-dependent manner, with IC_50_ values of about 1.5 μM (data not shown), which are within the range of clinical relevance. Direct counting for living cells by the trypan blue dye exclusion test confirmed that OTX015, used at 1 μM and 3 μM final concentration, was able to induce a significant reduction in SKOV3 cell number in comparison to mocked control cells (Figure 1a).

Furthermore, drug treatment clearly affected the morphological appearance of SKOV3, as confirmed by Giemsa staining, with OTX015-treated cells that became smaller, less dense, and showed longer filaments in comparison to DMSO-treated cells (Figure 1b). To further determine whether the OTX015-dependent reduction in OC cell growth was due alterations in cell cycle progression, flow cytometry analysis was performed in SKOV3 cells treated with or without OTX015 at different concentrations (0–1–3 μM). OTX015-treated cells showed a drastic arrest at G1 phase and a decrease in both S and G2 phases, whilst mocked control cells rapidly progressed through the cell cycle (Figure 1c). Consistent with the observed G1 arrest, an evident upregulation of p21 protein levels and its translocation in the nuclear compartment were found in OTX015-treated cells in comparison with DMSO-exposed controls (Figure 1d). Trans-well migration assays demonstrated that OTX015 treatment (1 μM and 3 μM) dose-dependently suppressed the ability of SKOV3 cells by about 30% (at 1 µM) and 50% (at 3 µM) to migrate through Boyden chamber membranes towards serum-containing medium when compared with DMSO control (Figure 1e). Similar results were obtained in UWB1.289 cells, as reported in Appendix A. Altogether, these results indicate that BET inhibition changes OC cell morphology, reduces proliferation through the induction of cell cycle arrest at G1 phase, and inhibits their migration potential.

### 3.2. OTX015 Induced Apoptosis by Modulating PTEN/AKT Pathway

In order to determine whether the reduction of cell viability was also related to cell death, we analyzed the presence of apoptotic cells by using the Annexin-V and 7-AAD staining in flow cytometry assays. As shown in Figure 2a, the percentage of SKOV3 cells undergoing apoptosis significantly increased in a dose-dependent manner after 72 h of OTX015 treatment (1 μM and 3 μM) compared to DMSO samples.

According with FACS results, OTX015 exposure altered the expression of apoptosis-related genes, such as cleaved PARP, BCL2 and active Caspase-3 proteins (Figure 2b). To deep insight the molecular pathways involved in OTX015-mediated apoptosis, we evaluated the PTEN/AKT signal pathway that is implicated in the proliferation of many cell types [36] and it is altered in several malignancies, including OC, with a marked downregulation of PTEN levels and a persistent activation of AKT protein [37]. We found that OTX015 exposure (0–1–3 μM) for 72 h was able to significantly increase PTEN protein levels, with a concomitant reduction of PTEN-downstream targets (Figure 2b). In particular, WB experiments showed a decreased phosphorylation status of AKT at serine-473 (p-AKT), whilst levels of total AKT protein were not affected by the specific drug treatment (Figure 2b). The re-expression of PTEN, by transiently transfecting SKOV3 cells with a full-length wild-type PTEN expression plasmid, coupled with OTX015 exposure (1 μM), resulted in a strong effect on as well as on cleaved PARP upregulation (Figure 2c), this suggesting that rescue of PTEN activity is one of the main molecular events linked to the OTX015-induced cell death in OC cells. A marked induction of PTEN levels in OTX015-treated cells and the concomitant downregulation of p-AKT were also observed in UWB1.289 cells (Appendix A), although they did not exhibit any evident apoptotic process neither alteration in both cleaved PARP and cleaved Caspase-3 protein expression (data not shown), this suggesting that BET inhibition induces a cytostatic effect more than a cytotoxic one in this OC cell line.

### 3.3. OTX015 Induces DNA Damage by Inhibiting NRF2-Mediated Antioxidant Mechanisms and Impacts on 3D Spheroid Architecture

As recently suggested, OTX015-mediated cell death seems to be correlated with DNA damage [11], so we analyzed the accumulation of activated H2AX histone (γH2AX), a well-established marker for DNA breaks [38], in SKOV3 cells treated with OTX015 (0–1–3 μM). WB assays demonstrated that the γH2AX levels were significantly elevated in OC treated cells in comparison to mocked controls at 72 h (Figure 3a, left panel). The strong accumulation of nuclear γH2AX foci was also demonstrated in IF experiments (Figure 3a, right panel), this confirming that the presence of DNA injury, that cells are unable to fix, has a primary role in the cellular lethality observed after OTX015 treatment.

Since DNA damage is often associated to a prominent production of reactive oxygen species (ROS), we assessed if BETi might be correlated with the NRF2-dependent antioxidant signaling pathway, which represents the main cellular defense mechanism against oxidative stress in eukaryotes. We demonstrated that BETi exposure was able to drastically downregulate NRF2 levels at both mRNA and protein levels by quantitative Real Time PCR (q-PCR), WB and IF assays (Figure 3b). Moreover, OTX015 treatment reduced the expression of specific NRF2-related downstream detoxifying enzymes, such as *GST-M1* and *GST-T1* genes (Figure 3c), this indicating that BETi blocked the molecular axis responsible of an active antioxidant defense.

Since BET inhibition has been recently found to be involved in the self-renewal capacity of cancer cells in different malignancies [39], we assessed the inhibitory effects of OTX015 on OC cell stemness features. OC cells were plated in ultra-low attachment 24-well plates with culture medium designed to allow spheroid formation for 48 h and then treated with or without 3 µM OTX015 for 10 days. Sphere-forming capacity was markedly perturbed by OTX015 treatment compared to vehicle, this suggesting that BETi was able to suppress self-renewal ability and in vitro tumorigenic capabilities of SKOV3 cells (Figure 3d, left panel). Similar results were obtained in UWB1.289 cells (Appendix A, left panel). Interestingly, protein expression analysis showed that CD133, a well-known stem cell surface marker, was significantly reduced by OTX015 exposure compared to mocked control cells in both SKOV3 cells (Figure 3d, right panel) and UWB1.289 samples (Appendix A, right panel), so speculating the anticancer effects of BETi on the cancer stem cell (CSC) population.

### 3.4. OTX015 Induces Nucleolar Stress by Downregulating GNL3 Expression in OC Cells

BETi has been reported to target BRD2/3/4 proteins by blocking the BRD pocket for the binding to acetylated histones and/or by directly reducing their expression. In rhabdomyosarcoma (RMS) cell lines, we recently demonstrated that OTX015 specifically downregulates *BRD4* and *MYC* levels, which in turn lead to a marked downmodulation of *GNL3* gene [11]. Interestingly, GNL3 silencing has been recently reported to have a role against OC tumorigenesis by reducing migration and epithelial-mesenchymal transition [30]. So, we analyzed the impact of OTX015 on *BRD4*, *MYC* and *GNL3* gene expression in OC cell lines. As reported in Figure 4a (left panel), exposure of SKOV3 cells to OTX015 was able to dose-dependently suppress *BRD4* mRNA levels, with a 50% and 60% reduction in 1 µM and 3 µM, respectively, in SKOV3 cells compared to untreated cells. Moreover, BRD4 protein was nearly absent in the cytoplasmic fraction and its expression was dramatically reduced in the nuclear compartment of OTX015-treated cells, as assessed by WB experiments performed on fractionated (nucleus-cytosol) protein extracts (Figure 4a, upper right panel). IF assays also demonstrated that BETi clearly decreased granular staining of BRD4 factor in the nuclear compartment (Figure 4a, lower right panel).

By q-PCR, WB and IF experiments, we also confirmed that OTX015-treated SKOV3 cells displayed a significant reduction of MYC mRNA and protein abundance in comparison to DMSO-treated samples (Figure 4b) as well as a drastic decrease of GNL3 mRNA and protein levels, already at the drug’s minor concentration (1 µM) (Figure 4b). Downregulation of GNL3 levels upon OTX015 exposure was also confirmed in UWB1.289 cells (Appendix A). Knocking down the expression of BRD4, the main target of OTX015 epi-drug, in SKOV3 cells by RNA interfering experiments with a pool of siRNAs (si-BRD4) against the *BRD4* mRNA, was able to cause a marked reduction of both MYC and GNL3 levels in comparison to si-NC control samples (Figure 4c), this reinforcing the perturbation of the BRD4/MYC/GNL3 molecular axis exerted by BETi.

### 3.5. OTX015 Treatment Improves the Efficacy of IR by Counteracting Radioresistance Mechanisms

To evaluate whether BETi might sensitize OC cells to ionizing radiation (IR), SKOV3 cells were treated with 1 µM OTX015 for 24 h, exposed to a 4 Gy of irradiation dose and used at different times upon IR. Clonogenic assays, performed by seeding cells at very low concentration in fresh medium at 4 h after irradiation and staining cells at 12 days after IR, confirmed that OTX015/IR cotreatment was significantly more efficient than BETi or to IR alone in diminishing colony formation capacity of SKOV3 cells (Figure 5a). Specifically, OTX015/IR coexposure clearly reduced the number of colonies of about 90% (*** *p* < 0.001), whilst single exposure with OTX015 or IR decreased colonies at about 50% (** *p* < 0.01) in comparison to DMSO-treated cells (Figure 5a, right panel). Markedly, colony reduction was significantly evident by comparing OTX015/IR vs. OTX015 alone or vs. IR (^##^
*p* < 0.01 and ^$$^
*p* < 0.01, respectively), this suggesting that OTX015 pre-treatment exerted enhanced radiosensitising effects on OC cells.

To deep insight the molecular mechanisms underlying the OTX015/IR enhanced antitumor effects on cell proliferation, survival and DNA damage, WB experiments were performed. The accumulation of p21 endorsed the induction of cell cycle arrest by OTX015 or IR alone, but maximum levels of this protein were observed in SKOV3 cells 24 h after the OTX015/IR coexposure (Figure 5b). The upregulation of γH2AX was synergistically induced by BETi and IR combined exposure (Figure 5b) despite the increased levels of both ATM and DNA-PK, two factors involved in DNA repair, in SKOV3 cells treated with OTX015 (with or without IR) in comparison to IR- or DMSO-exposed cells (Figure 5b). So, OTX015 might counteract the resistance mechanisms to radiotherapy by modulating DNA repair machinery-related factors, this being compatible with unfixed and accumulated DSBs that in turn induce cell death. In this context, IR alone did not upregulate the expression of both cleaved PARP and Caspase-3 proteins, whilst an additive accumulation of these apoptotic markers was observed in combination with OTX015 pre-treatment (Figure 5c). Interestingly, GNL3 mRNA and protein levels were significantly downregulated only in the presence of OTX015 treatment (Figure 5d) (with or without IR) in comparison to DMSO-treated cells; on the contrary, GNL3 levels were not perturbed by IR alone. In particular, GNL3 protein levels showed an enhanced drop-down when OTX015/IR combination was used in comparison to OTX015 alone or IR alone (Figure 5d, right panel). These data indicate that a 24 h pre-treatment with the BETi epi-drug can sensitize OC cells to IR, so counteracting potential radioresistance mechanisms. The clonogenic potential reduction (Appendix A) as well as the marked downregulation of GNL3 levels (Appendix A) after OTX015 exposure, as single agent or in combination with IR, were also confirmed in UWB1.289 cells. In all, these results suggest that a combined clinical approach using OTX015 and IR might efficiently radiosensitize cancer cells in order to potentiate radiotherapy-mediated antitumor effects in OC affected patients.

### 3.6. GNL3 Knocking-Down Potentiates the Antitumor Efficacy of BETi Treatment and Irradiation

To evaluate the contribution of GNL3 to the BETi-related anticancer properties, SKOV3 cells were transfected with a specific pool of siRNAs against *GNL3* mRNA (si-GNL3) or a control siRNA (si-NC), and after 72 h cells were counted and RNA/proteins were extracted. A significant downregulation of *GNL3* mRNA and protein levels (Figure 6a, left panel) was found in SKOV3 cells transfected with si-GNL3 compared to si-NC control cells. Moreover, WB analysis of DNA damage and apoptotic markers demonstrated that si-GNL3 transfection was able to increase γH2AX accumulation and cleaved PARP expression levels (Figure 6a, right panel).

Notably, when GNL3 silencing was used together with OTX015 (1 µM), the lowest effect on proliferation rate was observed compared to control cells (** *p* < 0.01 in Figure 6b). GNL3 siRNA drastically reduced the colony formation ability as shown in Figure 6c (left panel), also indicated by crystal violet absorbance in which comparisons between si-GNL3/OTX015 vs. si-GNL3 transfection alone or vs. OTX015 alone were significant (^#^
*p* < 0.05 and ^$^
*p* < 0.05, respectively in Figure 6c, right panel), this confirming a synergist effect of the cotreatment. Moreover, colony-forming assays confirmed that si-GNL3/IR coexposure was significantly more efficient than the single exposure to GNL3 siRNA transfection or to IR in dropping colony formation ability of SKOV3 cells (Figure 6d), with a reduced number of colonies to approximately 85% (*** *p* < 0.001) in si-GNL3/IR samples, whilst si-GNL3 led to a reduction at about 50% (*** *p* < 0.001) and IR of about 60% (*** *p* < 0.001) in comparison to DMSO-treated cells (Figure 6d, right panel). Markedly, comparisons between si-GNL3/IR vs. GNL3 siRNA or vs. IR alone were significant (^###^
*p* < 0.001 and ^$$^
*p* < 0.01, respectively), this indicating that GNL3 knocking down is able to radiosensitize OC cells.

### 3.7. Computational Analysis on GNL3 Confirms Its Upregulation in OC Patients and Predicts an Integrative Modellization of Its Biological Pathways

Significantly higher levels of *GNL3* mRNA were found by q-PCR experiments in our set of OC biopsies in comparison to normal ovarian tissues (3.6-fold change; *** *p* < 0.001), as shown in Figure 7a.

Important information on GNL3 expression in OC patients were also extrapolated by taking advantage of the free available R^2^ Genomics analysis and Visualization Platform (http://r2.amc.nl (accessed on 20 December 2020)). Indeed, MegaSampler interrogation confirmed the aberrant upregulation of *GNL3* gene across nine different OC datasets, for a total of 949 patients (Figure 7b). To infer the functional link between GNL3 protein and biological pathways perturbed by GNL3 knocking down in OC context, specifically DNA repair and apoptosis, a computational model based on network theory of GNL3 Network (GNL3_N) was realized (Material S1). Bioinformatic data were derived from different sources, such as genomic context, high-throughput experiments, conserved co-expression and previous knowledge. The topological analysis of the model showed that it has 1880 nodes linked by 4456 interactions and it is constituted by 37 connected components (data not shown), in which the main connected component (MC_GNL3_N) accounts for over than 85% of nodes (1599 on 1880). So, all the analyses were carried out on MC_GNL3_N. Network representation by Perfuse Directed Layout and results of the topological parameters are shown in Appendix A. The node size is proportional to the node connectivity, whereas the node color gradient is dependent on the clustering coefficient values (low: green to high: red). MC_GNL3_N conforms to the classical BA model of scale free networks [31], where the node degree follows the general power law: P(k)~k-γ, where: P = node degree function probability; k = node degree; γ = exponent. This is a very intriguing find because this specific topology, typical of signaling transduction chains [40,41], confers biologically relevant properties to the network and allows the identification of the most connected nodes, i.e., the hubs that are active in acting the strongest control over the network itself [42]. In our bioinformatic analysis, we identified 44 hubs (Figure 7c) and, in order to have a complete view of control mechanisms active in MC_GNL3_N, we examined a second class of nodes that exert a strong control, the so-called BottleNecks, which are mainly involved in driving information flow within the biological network. Indeed, we distinguished 44 BottleNecks and we were also able to recognize 28 nodes that were both hubs and BottleNecks (Figure 7c), with GNL3 representing one of these key controllers. As shown in Figure 7d, there are several subpopulations of controllers, depending on their values of connectivity and bottle neck score. The largest one is characterized by lower values of both the parameters, while the other one accounts for a lower number of nodes and higher values, in keeping with the scale free topology of the network. Interestingly, GNL3 is located between the two larger subpopulations, this strengthening its role as key player in the control of the biological functions represented by the model.

## 4. Discussion

Ovarian cancer (OC) is the fifth most frequent cause of cancer-related mortality in women, having the worst prognosis of all gynecological cancers with about 150,000 deaths a year. Despite worldwide research into this field and the overall cure rate has increased, the long-term prognosis for patients with metastatic and recurrent OC tumors remains dismal with few novel therapeutic options. Debulking surgery, chemotherapy and radiotherapy represent the standard treatments against OC tumors together with specific targeted therapies based on PARP inhibitors, but some patients do not positively respond to these therapies, showing progression or relapse of the disease due to the onset of resistant cancer stem cells (CSCs), so the need for effective clinical protocols based on combinatorial and more effective therapies, with a prominent impact against CSCs, is mandatory.

Epi-drugs have been tested in different tumors, including OC, in preclinical and in vivo studies and BET inhibition, based on a broad range of molecules (JQ1, OTX015, I-BET151, INCB057643, etc.) has been proposed as a promising therapeutic approach against OC progression, mainly in combination with conventional treatments. Anyway, it is essential to clearly define the specific molecular mechanisms exerted by BET inhibitors (BETi) in OC tumor type in order to thoroughly discover critical targets and novel biomarkers for selective personalized therapies and a faster diagnosis for women affected by this deadly malignancy.

In the present study, we more deeply investigated the antitumor effects of OTX015, an orally BETi molecule recently approved in different clinical trials on solid and hematological tumors, in OC cellular models, both as single agent and in combination with radiation therapy. OTX015 interferes with pivotal OC-related biological processes, such as proliferation, survival, migration and clonogenic ability, by rescuing the expression of the tumor suppressor gene *PTEN*, which in turn negatively regulates the PI3K/AKT axis, also in line with the molecular mechanism of action recently reported in other cancer types [37,43]. In particular, we found that OTX015 blocked cell cycle progression, with a G1/S arrest that correlated with a marked p21 accumulation, and it was able to induce apoptotic process as assessed by the BCL2 downregulation and the concomitant upregulation of both cleaved PARP and Caspase-3 proteins. Our results indicate the central role of the PTEN protein, which was found to be expressed at very low levels in OC cellular models, in apoptosis process by overexpressing a wild type PTEN vector in combination with OTX015 exposure, this exacerbating OC cellular death by inhibiting the AKT constitutive activation. The ability of OTX015, even at low concentration, to upregulate PTEN expression in OC cell lines is particularly interesting in the context of combined treatments since severe PTEN loss of function has been reported to be associated with resistance to specific targeted therapies, such as to those based on Trastuzumab [44]. Moreover, the sensitivity to OTX015 was also associated to DNA damage, as assessed by the strong and persistent upregulation of the histone H2AX phosphorylation at Serine 139, a known marker of genomic instability [38], this indicating the inability of cancer cells to repair DNA. Interestingly, also the NRF2 pathway, which is a major contributing player against oxidative stress, was impaired upon BETi exposure so increasing ROS production that can induce oxidative damage-mediated DNA repair and cell death. Finally, we also observed that OTX015 treatment strongly affected the architecture of tumor spheroids, which are a source of cancer stem-like cells involved in tumor proliferation, maintenance and spreading, and suppressed CD133 expression, this having a potential impact in overcoming drug resistance mechanisms and so improving the clinical outcome of OC patients. Indeed, different BETi were found to reduce CSC specific markers and stem cell-like features in both in vitro and in vivo experiments in different human malignancies, comprehending OC [21,39,45].

Moreover, the present data demonstrate that BET inhibition in combination with irradiation (IR) is able to enhance the radiosensitivity of OC cell lines. Resistance to conventional therapies, such as radiotherapy, represents an important factor that limits the success of antitumor treatments and contributes to patient poor survival. Specifically, our findings highlight the potential of OTX015 and IR coexposure in drastically reducing the colony-forming ability not only in comparison to untreated cells, but also to IR alone and to OTX015 alone, this clarifying that, when combined, the two treatments are additive. Also, the expression levels of different markers of DNA repair system, cell proliferation and apoptosis showed a more prominent modulation in OTX015/IR treated cells than upon single exposure. In line with reported studies on BETi action in different malignancies [11,15], OTX015 was able to impair BRD4 expression in OC cells, this causing a marked downregulation of *MYC* and *GNL3* mRNA and protein levels in both SKOV3 and UWB1.289 cells, as also confirmed by the specific knocking down of *BRD4* gene. The reduced expression of *GNL3* gene, encoding for nucleostemin, is particularly interesting in the context of OC tumors since this protein has been found to have a role in the migration ability and in the epithelial-mesenchymal transition of OC cells [30]. A dysregulated expression of GNL3 has been observed in several cancer cells and it was found to be associated with forced cell growth and survival, increased propensity to metastasize as well as with enhanced resistance to therapies, relapse and poor prognosis [25,26,27,28,29]. Interestingly, OTX015 and IR combination showed an enhanced effect in reducing the expression of GNL3 protein, this having a potential benefit in counteracting radioresistance mechanisms and so improving the therapeutic efficacy against OC tumors. By using RNA interference methodology, we demonstrated that GNL3 knocking down interfered with specific cellular features related to clonogenic potential and survival. Accordingly, downregulation of GNL3 expression was able to significantly induce apoptosis in leukemia cellular models [46]. Since inefficient apoptosis represents one of the primary mechanisms that lead to drug resistance, GNL3 reduction may likely be a molecular target for OC treatment by reducing tumor growth and dissemination. The oncogenic role of *GNL3* gene in OC tumors is also supported by the significant upregulation found in our cohort of OC patient’s biopsies in comparison to normal ovarian tissues. Moreover, our integrated analysis of gene expression data in ovarian tumors from publicly available databases led us to identify that *GNL3* mRNA is aberrantly upregulated in a wide number of OC patients. The association between excessive levels of GNL3 and more aggressive tumor disease was also reported in the study by Lin et al. [29], in which high GNL3 expression by immunohistochemistry staining positively correlated with advanced stage and histological grade of OC tumors. This correlation suggests that GNL3 has the potential to be a prognostic marker and underlines the importance of considering GNL3 evaluation in the clinical management of women with OC. Interestingly, our bioinformatic analyses underline that GNL3 is one of the central nodes of the molecular network of protein interactions that are involved in different processes and functions linked to OC tumorigenesis. In our bioinformatic model, we identified 44 hubs (less than 3% of the total nodes) meaning that we have only 3% probability to target a hub, when randomly attacking the network. Clearly, in one hand this topology makes BA networks robust against random attacks, and on the other one hand makes them susceptible to targeted control [31,32,47]. The large majority of nodes within the network are scarcely linked, consequently in the most of cases a random perturbation will affect actually those nodes, with no or negligible effects on network topology (i.e., on biological function). We also identified different BottleNeck nodes, that represent a parameter that is a measure of the node centrality, expressed as the number of shortest paths in which the node is involved, and it is related to the importance of nodes in controlling the information flux within directed network [42]. As already noted, the whole system can be modulated with high efficiency by controlling only a small number of molecules, reducing in this way the energetic cost and facilitating or accelerating the cell response. The identification of GNL3 among the small numbers of hubs and BottleNecks of the network could be helpful to controlling the biological systems with the purpose of develop diagnostic/therapeutic strategies based on the pharmacological control of this central regulator. Future studies will be addressed to translate OTX015-, si-GNL3- and IR-based experiments in xenograft mice with the final aim to set more effective and less toxic protocols for OC treatment.

In conclusion, OTX015 treatment is likely to represent an effective and widely applicable strategy against OC by interfering with cell proliferation and survival through GNL3 downregulation. The concomitant treatment with BETi and radiotherapy might provide an additional therapeutic approach to increase the efficacy of antitumor treatments and, consequently, survival of OC patients and a deeper insight into the molecular mechanisms underlying tumorigenesis of this aggressive female tumor.

## 5. Conclusions

OTX015 significantly exerts antitumor effects against OC cells by downmodulating GNL3 expression. This epi-drug represents a promising therapeutic approach in OC treatment and supports the possibility to combine BETi with radiotherapy in order to induce highest cytotoxicity against ovarian tumor cells and to impair possible radioresistance-related mechanisms.

## Figures and Tables

**Figure 1 cancers-13-01519-f001:**
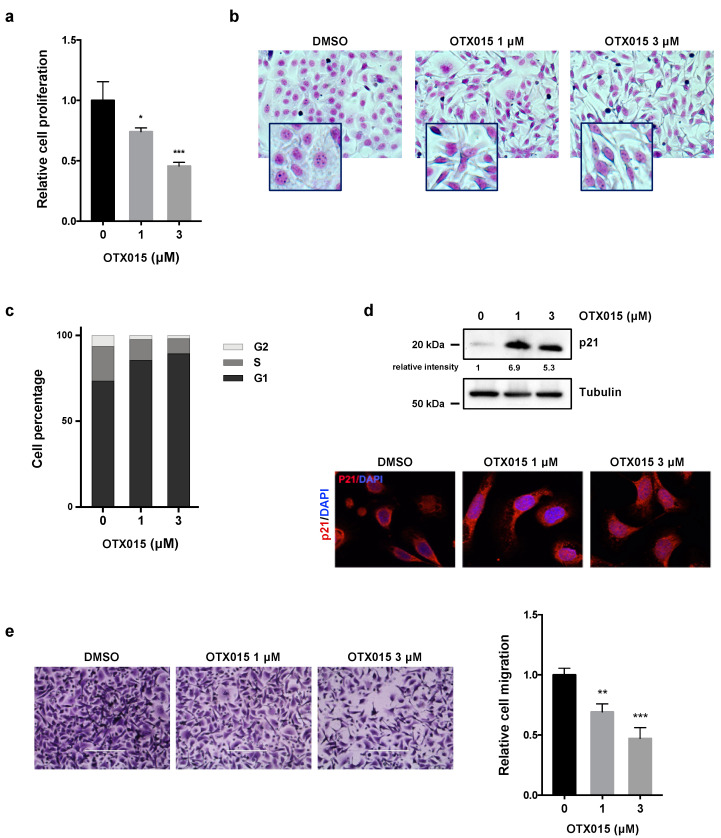
OTX015 influences viability, morphology, proliferation and migration of OC cells. (**a**) SKOV3 cell proliferation, evaluated by trypan blue exclusion dye at 72 h after OTX015 treatment (1–3 μM), was graphed as fold increase over control (0 μM), set at 1. Results represent mean values ± SD of three independent experiments. Statistical analyses were performed by using one-way ANOVA: * *p* < 0.05; *** *p* < 0.001 vs. not-treated cells (**b**) Cellular morphology, evaluated by Giemsa staining, of SKOV3 cells 72 h after OTX015 exposure at 1 μM or 3 μM concentration. Control cells were treated with DMSO. (**c**) Flow-cytometry experiments showing cell percentage at G1, S and G2 phases in SKOV3 cells treated for 48 h with OTX015 (0–1–3 μM). Data are average value of two independent experiments. (**d**) WB assays (upper panel) of p21 protein levels in OTX015-exposed SKOV3 cells (0–1–3 μM). Tubulin served as loading control. IF staining (lower panel) of p21 protein in SKOV3 cells exposed to OTX015 (1 μM and 3 μM). Control cells were treated with DMSO. DAPI was used for nuclear staining. Images were captured under ApoTome microscope at 40× magnification. (**e**) Cell migration ability in OTX015-exposed SKOV3 cells (1 μM and 3 μM). Representative photos of migrated cells stained with crystal violet (magnification of 20×). Bars are the average values ± SD of migrated cells in two independent experiments, each in triplicate. One-way ANOVA was used for statistics: ** *p* < 0.01; *** *p*-value < 0.001 vs. DMSO negative control. The original Western Blot images can be found in Appendix A.

**Figure 2 cancers-13-01519-f002:**
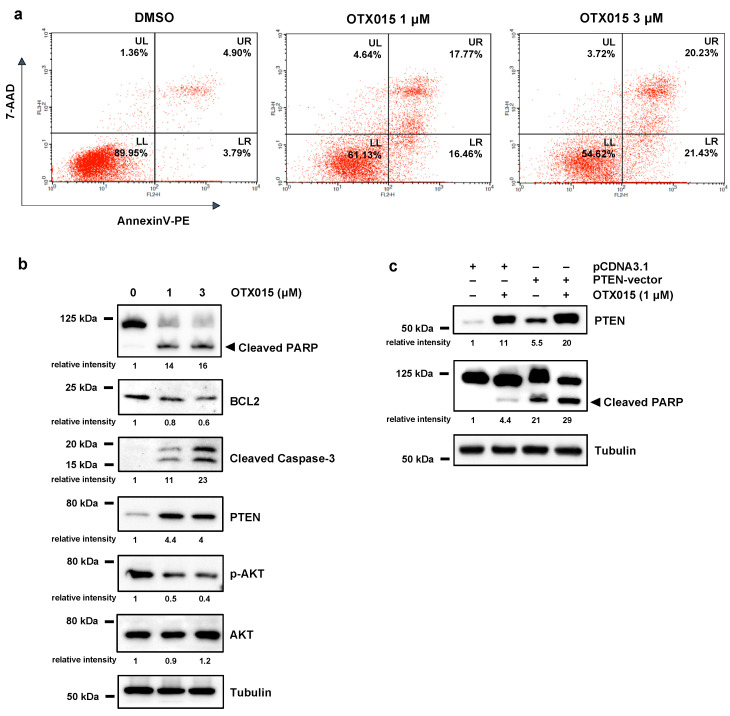
BET inhibition induces apoptosis by modulating PTEN/AKT. (**a**) Flow-cytometry assays by Annexin V/7AAD staining show cell death rates in OTX015-exposed SKOV3 cells (1 μM and 3 μM) and DMSO controls. Representative dot plots with mean values of two independent experiments showing the percentage of viable cells (quadrant LL, Annexin V−/7AAD−), early apoptotic cells (quadrant LR, Annexin V+/7AAD−), late apoptotic cells (quadrant UR, Annexin V+/7AAD+) and necrotic cells (quadrant UL, Annexin V−/7AAD+). (**b**) WB analysis of the apoptotic markers cleaved PARP, BCL2 and Caspase-3 in OTX015-exposed SKOV3 cells (0–1–3 μM) for 72 h. Protein analysis showing the strong PTEN increase and the downregulation of phospho (p)-AKT levels in OTX015-exposed SKOV3 (0–1–3 μM) for 72 h. Tubulin served as loading control and representative blot was shown. (**c**) WB experiments showing PTEN and cleaved PARP expression in SKOV3 cells transfected with PTEN-vector or pcDNA3.1 empty vector and treated with or without 1 μM OTX015 for 72 h. Tubulin served as loading control and representative blot was shown. The original Western Blot images can be found in Appendix A.

**Figure 3 cancers-13-01519-f003:**
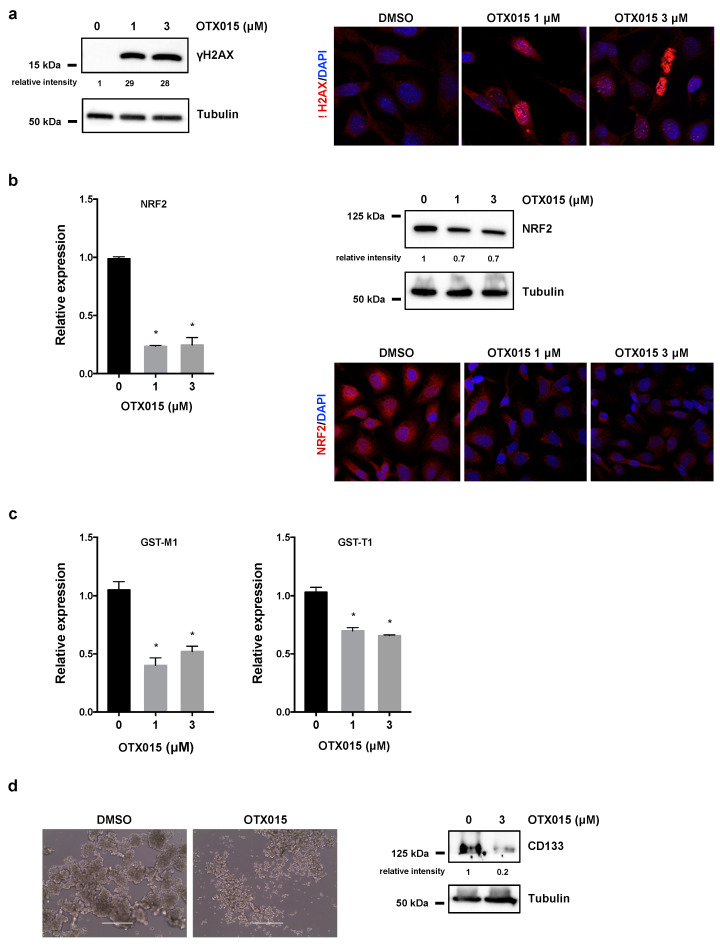
BET inhibition induces DNA damage and perturbs tumor spheroid architecture. (**a**) WB analysis (left panel) of γH2AX in SKOV3 cells treated with OTX015 (0–1–3 μM) for 72 h. Tubulin served as internal control. IF experiments (right panel) showing the presence of γH2AX foci (red) in OTX015 treated (1 and 3 μM) cells. DMSO was used in control cells. Nuclei (blue) were visualized with DAPI staining. Representative images captured under ApoTome microscope at 40× magnification. (**b**) q-PCR analysis of *NRF2* mRNA levels (left panel) in SKOV3 cells exposed to OTX015 (0–1–3 μM), graphed as fold increase over not-treated cells, set at 1. *β-actin* mRNA was used as endogenous control. Histograms indicate mean values ±SD of two independent experiments, each performed in triplicate. Statistical significance vs. control cells was calculated by one-way ANOVA (* *p* < 0.05). WB experiments (upper right panel) performed on protein extracts from SKOV3 cells treated with OTX015 (0–1–3 μM) for 72 h. Tubulin served as loading control. IF experiments (lower right panel) showing the downregulation of NRF2 protein expression (red) in OTX015-exposed (1 μM and 3 μM) cells respect to untreated SKOV3 samples. (**c**) Transcript levels of *GST-M1* and *GST-T1* genes by q-PCR analysis in SKOV3 cells exposed to OTX015 (0–1–3 μM) and graphed as fold increase over not-treated cells, set at 1. *β-actin* mRNA was used as endogenous control. Histograms indicate mean values ±SD of two independent experiments, each performed in triplicate. Statistical significance vs. control cells was calculated by one-way ANOVA (* *p* < 0.05). (**d**) OTX015 alters spheroid formation in SKOV3 cells cultured in 24-well ultralow attachment plates with DMEM-F12/B27/EGF/FGF medium. Cells were photographed under a light microscope at 20× magnification (left panel) after 10 days of 3 μM OTX015 treatment (DMSO as control). WB of CD133 protein levels (right panel) in OTX015-exposed (0–3 μM) spheroids. Tubulin served as loading control. The original Western Blot images can be found in Appendix A.

**Figure 4 cancers-13-01519-f004:**
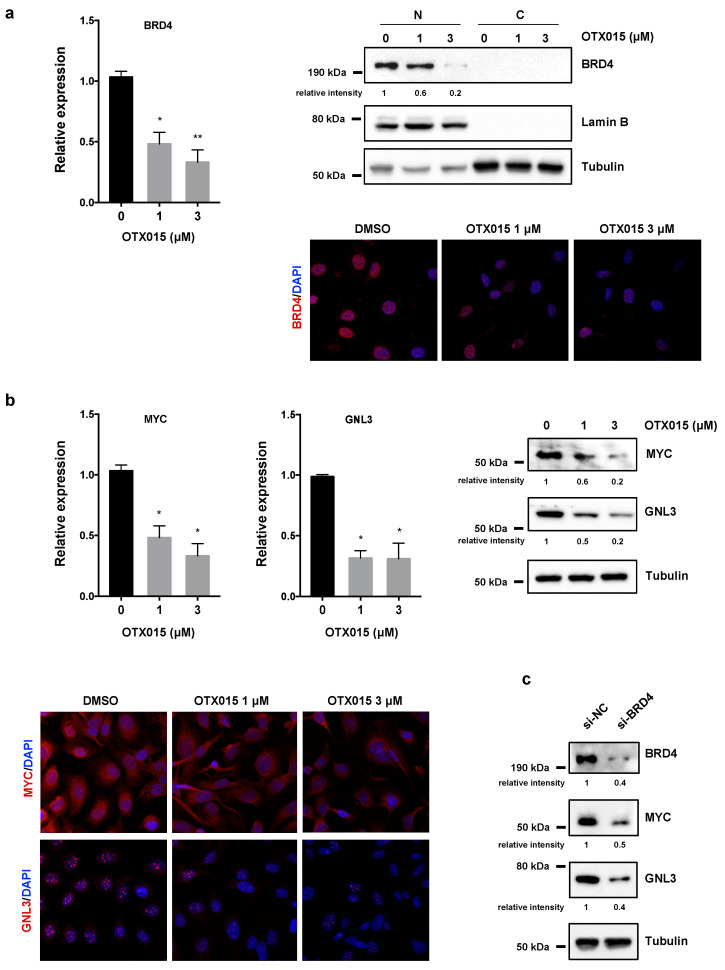
Effects of OTX015 on BRD4, MYC and GNL3 expression in OC cells. (**a**) Experiments of q-PCR (left panel) showing *BRD4* mRNA levels in OTX015-exposed SKOV3 cells (0–1–3 μM), expressed as fold change relative to control cells, set at 1. *β-actin* mRNA was used as endogenous control. Histograms indicate mean values ± SD of two independent experiments, each performed in triplicate. Statistical significances were calculated by one-way ANOVA (* *p* < 0.05; ** *p* < 0.01). WB analysis (upper right panel) performed on nuclear (N) and cytoplasmic (C) protein extracts from OTX015-exposed SKOV3 cells (0–1–3 μM) for 72 h. Lamin B served to normalize nuclear fraction, whilst tubulin was used for the cytoplasmic fraction. IF experiments (lower right panel) showing the reduction of BRD4 nuclear staining (red) in OTX015 treated (1 μM and 3 μM) cells respect to DMSO controls. DAPI (blue) for nuclear staining. Representative images captured under ApoTome microscope at 40× magnification. (**b**) q-PCR analysis (left panel) of *MYC* and *GNL3* expression in OTX015-exposed SKOV3 cells (0–1–3 μM), each expressed as fold change relative to mocked control cells, set at 1. *β-actin* mRNA was used as endogenous control. Histograms indicate mean values ± SD of two independent experiments, each performed in triplicate. Statistical significances were calculated by one-way ANOVA (* *p* < 0.05 vs. mocked control). WB analysis (right panel) performed on total protein extracts from SKOV3 cells treated with OTX015 (0–1–3 μM) for 72 h. Tubulin served as loading control and representative blot was shown. IF experiments (lower left panel) showing the downregulation of MYC expression and the lower intensity of GNL3 nucleolar foci in OTX015 treated (1 μM and 3 μM) cells in comparison to DMSO controls. (**c**) WB analysis of BRD4, MYC and GNL3 proteins performed on total protein extracts from SKOV3 cells transfected for 72 h with BRD4 siRNAs (si-BRD4) or negative control molecules (si-NC). Tubulin served as loading control and representative blot was shown. The original Western Blot images can be found in Appendix A.

**Figure 5 cancers-13-01519-f005:**
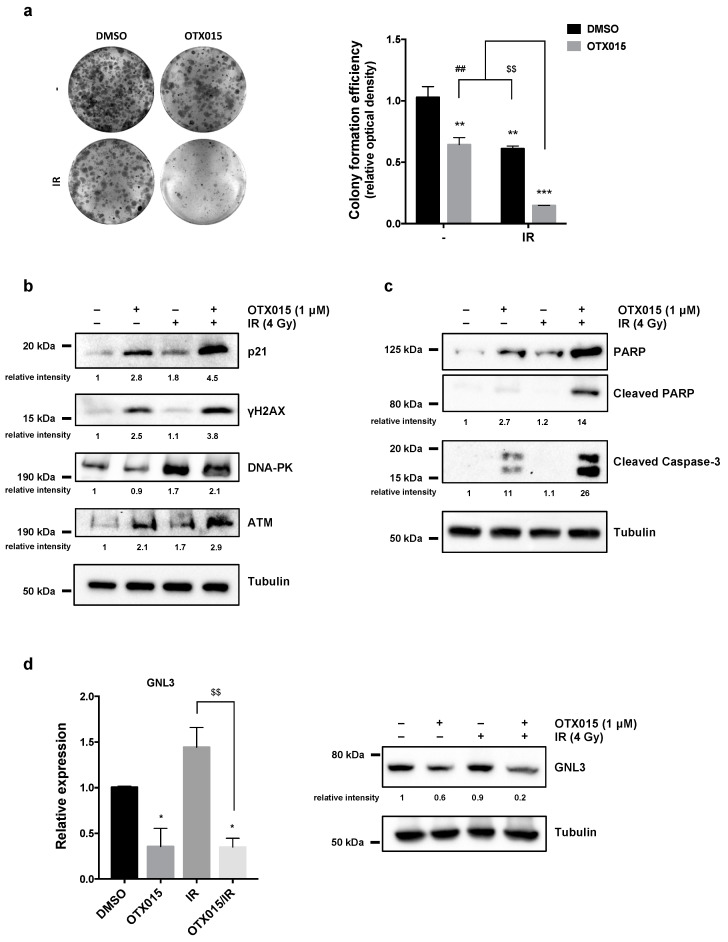
OTX015 exposure radiosensitizes OC cells by inducing apoptosis and inhibiting colony-forming ability and DNA damage repair. SKOV3 cells, treated with or without OTX015 (1 μM), were exposed or not to radiation (4 Gy). (**a**) Four h after IR, SKOV3 cells were plated at low concentration and cultured for 12 days. Representative images of colonies marked with crystal violet. Colony formation efficiency was obtained by crystal violet optical density from different assays, each in triplicate. Histograms are means ± SD. Statistical significances were calculated by two-way ANOVA: ** *p* < 0.01 and *** *p* < 0.001, vs. DMSO without IR; ^$$^
*p* < 0.01 vs. DMSO/IR; ^##^
*p* < 0.01 vs. OTX015 without IR. (**b**) WB of selected markers of cell cycle arrest (p21) and DNA damage/response (γH2AX, DNA-PK, ATM) in SKOV3 cells with or without OTX015 and/or IR. Tubulin served as normalizer and representative blot was shown. (**c**) WB of specific proteins involved in apoptosis (PARP, Caspase-3) were performed on SKOV3 cells treated with or without OTX015 and/or IR. Tubulin served as loading control and representative blot was shown. (**d**) q-PCR analysis (left panel) of *GNL3* transcript levels in OTX015/IR-exposed SKOV3 cells, expressed as fold change over control cells (DMSO), set at 1. *β-actin* mRNA was used as endogenous control. Histograms indicate mean values ± SD of three independent experiments, each in triplicate. Statistical significances were calculated by two-way ANOVA (* *p* < 0.05 vs. DMSO; ^$$^
*p* < 0.01 vs. IR). WB analysis (right panel) performed on protein extracts from OTX015/IR-exposed SKOV3 cells. Tubulin served as normalizer. The original Western Blot images can be found in Appendix A.

**Figure 6 cancers-13-01519-f006:**
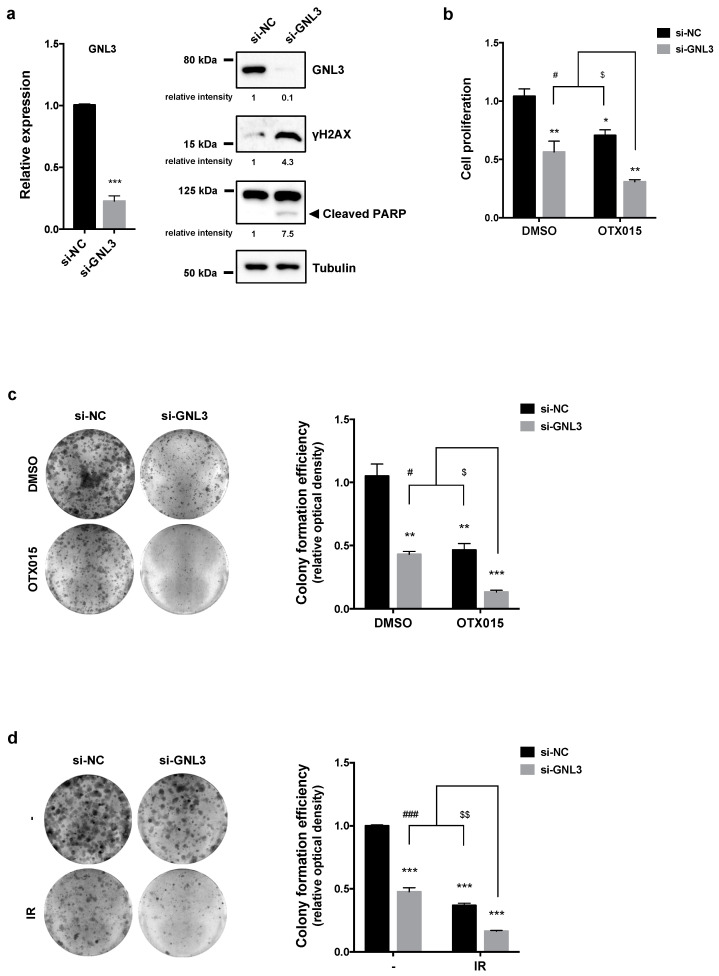
Effects of GNL3 knocking down in OC cells. (**a**) q-PCR analysis (left panel) of *GNL3* transcript levels in SKOV3 cells transfected for 72 h with GNL3 siRNAs (si-GNL3), expressed as fold change over mocked control cells (si-NC), set at 1. *β-actin* mRNA was used as endogenous control. WB analysis (right panel) of GNL3, γH2AX and PARP proteins performed on SKOV3 cells transfected for 72 h with GNL3 siRNAs (si-GNL3) or negative control molecules (si-NC). Tubulin was used as loading control and representative blot was shown. (**b**) Cell proliferation, evaluated by trypan blue exclusion dye in SKOV3 cells transfected with or without si-GNL3 and exposed to 1 μM OTX015, was expressed as fold change respect to si-NC/DMSO sample, set at 1. Results represent mean values ± SD of two independent experiments. Statistical significances were calculated by two-way ANOVA: * *p* < 0.05, ** *p* < 0.01; *** *p* < 0.001 vs. si-NC/DMSO; ^#^
*p* < 0.05 vs. si-GNL3/DMSO; ^$^
*p* < 0.05 vs. si-NC/OTX015. (**c**) Clonogenic assays after 12 days of culture in SKOV3 cells transfected with or without si-GNL3 and treated with 1 μM OTX015. Representative images of colonies marked with crystal violet and colony formation efficiency obtained by crystal violet absorbance. Experiments were carried out twice, each in triplicate. Histograms are means ± SD. Statistical significances were calculated by two-way ANOVA: ** *p* < 0.01, *** *p* < 0.001, vs. si-NC/DMSO; ^#^
*p* < 0.05 vs. si-GNL3/DMSO; ^$^
*p* < 0.05 vs. si-NC/OTX015. (**d**) Four h after IR, si-NC and si-GNL3 transfected SKOV3 cells were plated at low concentration and cultured for 12 days. Representative images of colonies marked with crystal violet, from two independent experiments, each performed in triplicate. Bar represents the means ± SD of the crystal violet absorbance. Statistical analyses were calculated by using two-way ANOVA: *** *p* < 0.001, vs. si-NC/no IR; ^###^
*p* < 0.001 vs. si-GNL3/no IR; ^$$^
*p* < 0.01 vs. si-NC/IR. The original Western Blot images can be found in Appendix A.

**Figure 7 cancers-13-01519-f007:**
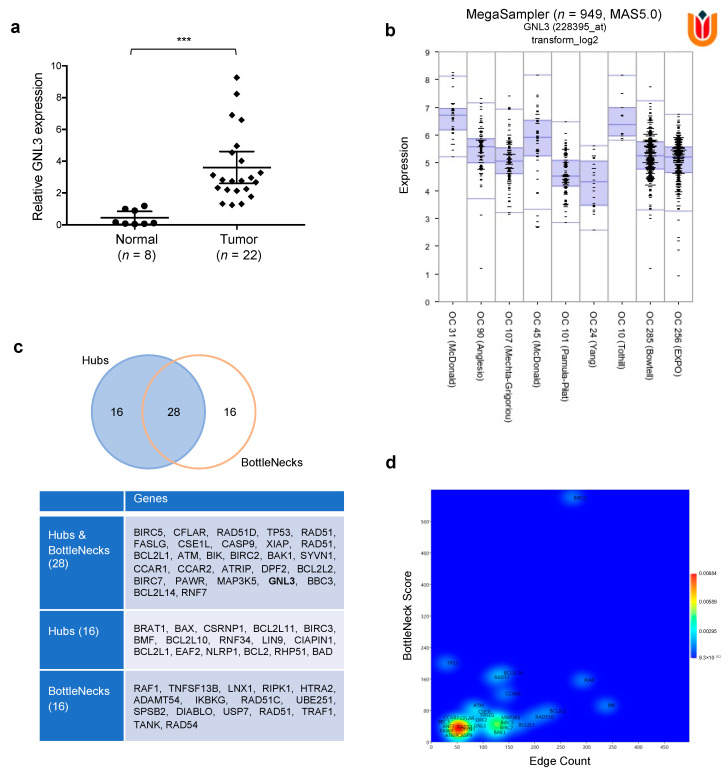
Biological significance of GNL3 expression in OC tumorigenesis. (**a**) *GNL3* mRNA expression was assessed in 22 OC tumor biopsies and 8 non-tumoral ovarian tissues by q-PCR assays (*** *p* < 0.001). (**b**) Using ‘R2: Genomics Analysis and Visualization Platform (http://r2.amc.nl (accessed on 20 December 2020)), the expression of GNL3 gene was assessed in 9 different databases referred to different Ovarian Cancers, and it results always overexpressed, *n* = 949, *p* = 2.2 × 10^−43^ (**c**) The nodes acting as controller within the network are identified and classified as Hubs and BottleNecks, Hubs, or BottleNecks depending on their node degree and BottleNeck score. (**d**) A 2D Kernel Density Estimation (2D-KDE), Gaussian model, was carried out on Hubs and BottleNecks based on their node degree (*x*-axis) and BottleNecks score (*y*-axis).

## Data Availability

Data that support the findings of this study are available from the corresponding author upon reasonable request.

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
