# Peer review of "OTX015 Epi-Drug Exerts Antitumor Effects in Ovarian Cancer Cells by Blocking GNL3-Mediated Radioresistance Mechanisms: Cellular, Molecular and Computational Evidence"

_cancers, 2021, doi:10.3390/cancers13071519_

Round 1
Reviewer 1 Report
This study deals with the effect of OTX-015 on ovarian carcinoma cell lines. There are several points to be addressed further.
#1. In Fig. 3d, the decrease of colony-forming capacity of tumor cells is not evident. A colony-forming assay or limiting dilution assay should be supplemented.
#2. In Fig. 5d, the combination has more effect than IR alone, but it is not clear that OTX-015 and IR have a true synergistic effect. OTX-015 alone seems to result in the similar outcome. Is there any evidence of radio-sensitization of OTX-015?
#3. The first line treatment for ovarian cancer is the combination of cisplatin and paclitaxel. The effect of any novel agent needs to be compared with the standard.
#4. OTX-015 is known as an epigenetic regulator. Is there any proof that wide-spread changes of epigenetic modulation take place after drug treatment? Only decrease of BRD4 expression is too weak to demonstrate the mechanism.
#5. No in vivo experiment is done. The effect of OTX-015 requires to be verified in animal models of ovarian cancer.
Reviewer 2 Report
THE CONTRIBUTION IS VERY INTERESTING EVEN MORE THAN 20 PAGES
Reviewer 3 Report
Thank you for allowing me to review this interesting manuscript.
The paper is well written
The topic is innovative, acutally I have not major comment
Just a minor commnet
Is there a correlation between OTX015 and BRCA status?
